# Taming Rectified Flow for Inversion and Editing

**Jiangshan Wang** [* 1 2]  **Junfu Pu** [* † 2]  **Zhongang Qi** [2]  **Jiayi Guo** [1]  **Yue Ma** [3]
**Nisha Huang** [1]  **Yuxin Chen** [2]  **Xiu Li** [1]  **Ying Shan** [2]

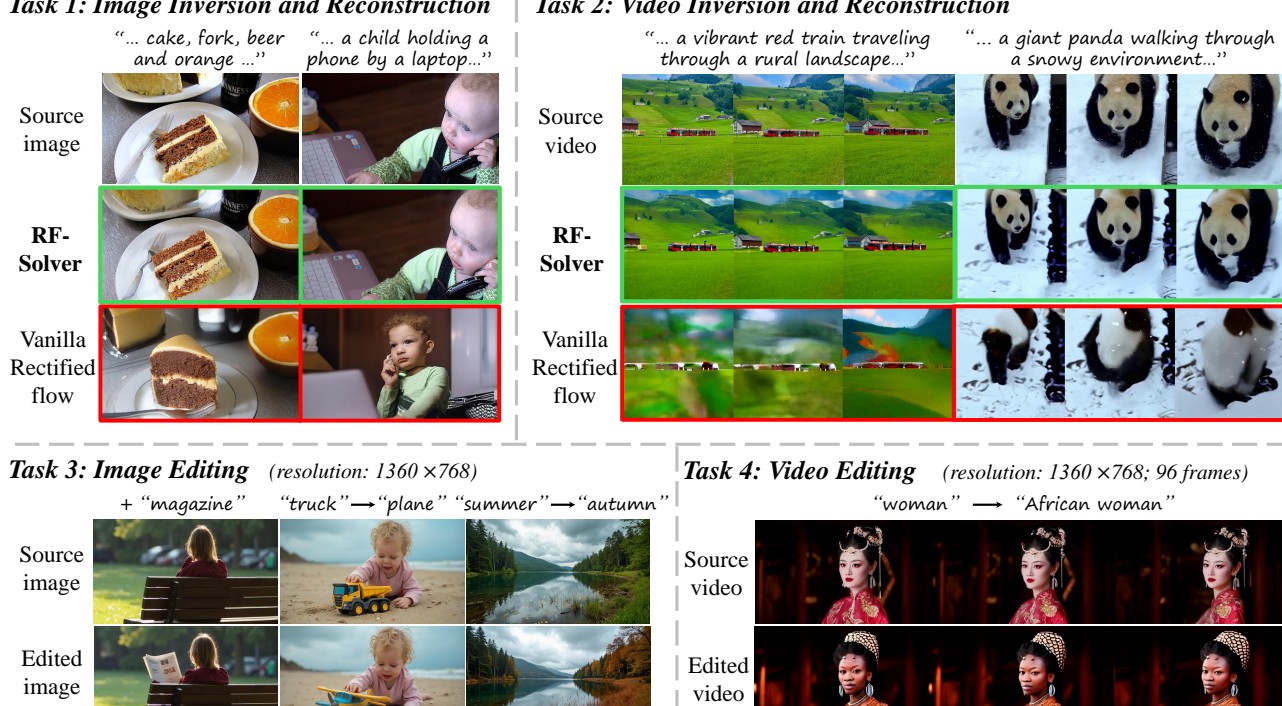

*Figure 1.* **RF-Solver for downstream tasks in image and video.** We propose **RF-Solver** to solve the rectified flow ODE with reduced error, thereby enhancing both sampling quality and inversion-reconstruction accuracy for rectified-flow-based generative models (Black-Forest-Labs, 2024; Zheng et al., 2024). Furthermore, we propose **RF-Edit**, which utilizes the RF-Solver for editing. Our methods demonstrate impressive performance across generation, inversion, and editing tasks in both image and video modalities.

## Abstract

Rectified-flow-based diffusion transformers like FLUX and OpenSora have demonstrated outstanding performance in the field of image and video generation. Despite their robust generative capabilities, these models often struggle with inversion inaccuracies, which could further limit their effectiveness in downstream tasks such as image and video editing. To address this issue, we propose RF-Solver, a novel training-free sampler that effectively enhances inversion precision by mitigating the errors in the ODE-solving process of rectified flow. Specifically, we derive the exact formulation of the rectified flow ODE and apply the high-order Taylor expansion to estimate its nonlinear components, significantly enhancing the precision of ODE solutions at each timestep. Building upon RF-Solver, we further propose RF-Edit, a general feature-sharing-based framework for image and video editing. By incorporating self-attention features from the inversion process into the editing process, RF-Edit effectively preserves the structural information of the source image or video while achieving high-quality editing results. Our approach is compatible with

[*]Equal contribution  [1]Tsinghua University [2]ARC Lab, Tencent PCG [3]HKUST. [†]Project lead: Junfu Pu <jevinpu@tencent.com>. Correspondence to: Zhongang Qi <zhongangqi@gmail.com>, Xiu Li <li.xiu@sz.tsinghua.edu.cn>.

*Proceedings of the 42nd International Conference on Machine Learning*, Vancouver, Canada. PMLR 267, 2025. Copyright 2025 by the author(s).

any pre-trained rectified-flow-based models for image and video tasks, requiring no additional training or optimization. Extensive experiments across generation, inversion, and editing tasks in both image and video modalities demonstrate the superiority and versatility of our method. Code is available at this URL .

## 1. Introduction

Recent advancements of generation methods based on Rectified Flow (RF) (Liu et al., 2022a; Lee et al., 2024; Wang et al., 2024b) have demonstrated exceptional performance in synthesizing high-quality images and videos. Different from traditional approaches represented by Stable Diffusion (Ho et al., 2020; Rombach et al., 2022), these methods leverage the Diffusion Transformer (Peebles & Xie, 2023; Yang et al., 2024c; Xie et al., 2024; Tang et al., 2024) architecture and implement a straight-line motion system to produce the desired data distribution. With these effective designs, FLUX (Black-Forest-Labs, 2024) and OpenSora (Zheng et al., 2024) have respectively emerged as one of the state-of-the-art (SOTA) methods in the field of Text-to-Image (T2I) and Text-to-Video (T2V) generation.

Despite the remarkable success in the fundamental T2I and T2V generation tasks, few studies have explored the performance of RF-based models on various downstream tasks such as inversion-reconstruction (Song et al., 2021a; Mokady et al., 2023; Guo et al., 2024; Wang et al., 2023c) and editing (Hertz et al., 2022; Meng et al., 2022). When directly applying the vanilla RF for inversion, we observe that it fails to faithfully reconstruct the image or video from the source. Examples are shown in Figure 1 Task 1 and Task 2 (the third row). For image inversion, the positions of objects (e.g., the cake) and the appearance of individuals (e.g., the child) in the reconstructed image significantly diverge from the source image. The performance of video inversion is even worse, with noticeable distortions present in the reconstructed video. The inaccuracies of inversion and reconstruction would severely constrain the performance of RF models on other inversion-based downstream tasks such as image editing (Hertz et al., 2022; Tumanyan et al., 2023; Nguyen et al., 2024; Duan et al., 2024; Ju et al., 2024) and video editing (Liu et al., 2024; Ku et al., 2024; Fan et al., 2024; Shin et al., 2024).

In this work, we investigate the aforementioned problem by delving into the inversion and reconstruction process of the RF. Specifically, we track the latent at each intermediate timestep during inversion and reconstruction, calculating the Mean Square Error (MSE) between them at corresponding timesteps. We observe that significant errors are introduced at each timestep throughout the whole reconstruction process, and their accumulation ultimately results in a consid-

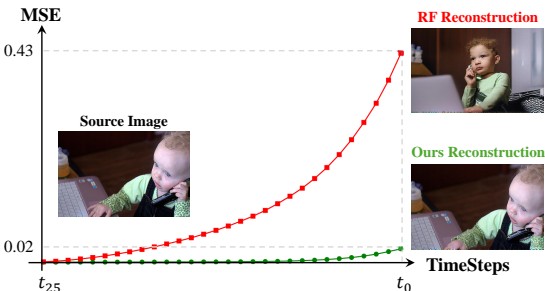

*Figure 2.* **Analysis of the inversion-reconstruction process.** Inversion takes the source image latent $\widetilde{\boldsymbol{Z}}_{t_0}$ as the input and progressively add noise for $N$ timesteps, obtaining $\widetilde{\boldsymbol{Z}}_{t_N} \in \mathcal{N}(0, \boldsymbol{I})$. $\widetilde{\boldsymbol{Z}}_{t_N}$ is then denoised for $N$ timesteps to obtain the reconstruction $\boldsymbol{Z}_{t_0}$. During this process, we store the latent $\widetilde{\boldsymbol{Z}}_{t_i}$ and $\boldsymbol{Z}_{t_i}$ at each timestep respectively in inversion and denoising processes. Then we calculate the Mean Squared Error (MSE) between them. The red curve represents the vanilla Rectified Flow inversion and the green curve represents RF-Solver inversion.

erably deviated output (the red curve in Figure 2). Based on the definition and inference process of RF (Liu, 2022; Liu et al., 2022a), we identify that these errors stem from the Ordinary Differential Equation (ODE) solving process (Zhang et al., 2024; Wang et al., 2024a; Hong et al., 2024; Peng et al., 2024). Specifically, the essence of the inversion and generation process for RF is to derive the solution of RF ODE (Liu et al., 2022a). Since this ODE includes terms involving complex neural networks, the solution can only be coarsely approximated by a sampler. However, the experiment in Figure 2 indicates that the sampler adopted in existing models (Black-Forest-Labs, 2024; Zheng et al., 2024) lacks sufficient precision for the inversion task, causing notable errors to accumulate at each timestep, finally leading to unsatisfactory reconstruction results.

Based on the analysis, we aim to improve inversion accuracy by introducing a more effective sampler, which is more general and fundamental than designing a specific inversion method. To this end, we propose **RF-Solver**. Specifically, we note that the exact formulation of the RF ODE solution can be directly derived using the variation of constants method. For the nonlinear component of this solution (i.e., the integral of the neural network), we utilize Taylor expansion for estimation. By employing higher-order Taylor expansion, the ODE can be solved with reduced error, thereby enhancing the performance of RF models. RF-Solver is a generic sampler that can be seamlessly integrated into any rectified flow model without additional training or optimization. Experimental results demonstrate that RF-Solver not only significantly enhances the accuracy of inversion and reconstruction (the green curve in Figure 2), but also improves performance on fundamental tasks such as T2I generation.

Building upon this, we propose **RF-Edit** to leverage RF-Solver in editing tasks. Real-world image and video editing require the model to make precise modifications to a

source image/video while maintaining its overall structure unchanged, presenting greater challenges than reconstruction. In this scenario, it is inadequate to solely rely on the inverted noises as prior knowledge for editing, which could lead to edited results being excessively influenced by the target prompt, diverging significantly from the original source (Hertz et al., 2022; Tumanyan et al., 2023). Addressing this problem, RF-Edit stores the $\mathcal{V}$ (value) feature in the self-attention layers at several timesteps during inversion. These features are used to replace the corresponding features in the denoising process. Practically, we design two specific sub-modules for RF-Edit, respectively leveraging the DiT structure of FLUX (Black-Forest-Labs, 2024) and OpenSora (Zheng et al., 2024) as the backbones for image and video editing. With the effective design of RF-Edit, it demonstrates superior performance in both image and video domains, outperforming various SOTA methods.

## 2. Related Work

### 2.1. Inversion

Inversion maps the real visual data, i.e., image and video, to representations in noise space, which is the reverse process of generation. The representative method, DDIM inversion (Song et al., 2021a;b), adds predicted noise recursively at each forward step. Many efforts (Elarabawy et al., 2022; Wallace et al., 2023; Mokady et al., 2023; Rout et al., 2024a; Miyake et al., 2023; Lu et al., 2022) have been made to mitigate the discretization error in DDIM inversion. Despite the effectiveness of inversion in diffusion models, the exploration of inversion in SOTA rectified flow models like FLUX and OpenSora is limited. RF-prior (Yang et al., 2024b) uses the score distillation to invert the image while it requires many optimizing steps. More recently, (Rout et al., 2024b) introduces an additional vector field conditioned on the source image to improve the inversion. However, the error from the original vector field of rectified flow still persists, which would limit the performance of such method on various downstream tasks. In contrast, we aim to directly mitigate the error from the original vector field in this work.

### 2.2. Image and Video Editing

Training-free methods for image and video editing (Huang et al., 2024a; Sun et al., 2024) have gained increasing popularity for their efficiency and effectiveness. Existing image editing methods focus on prompt refinement (Ravi et al., 2023; Wang et al., 2023a), attention-sharing mechanism (Hertz et al., 2022; Parmar et al., 2023; Cao et al., 2023; Tumanyan et al., 2023), mask guidance (Avrahami et al., 2023; Couairon et al., 2022; Huang et al., 2023), and noise initialization (Brack et al., 2023; Yang et al., 2023b). Video editing introduces additional complexities in maintaining temporal consistency, making it a more challenging task. Existing video editing methods focus on attention injection

(Qi et al., 2023; Wang et al., 2023b; Liu et al., 2024), motion guidance (Cong et al., 2023; Geyer et al., 2023; Yang et al., 2024a; Wang et al., 2024c), latent manipulation (Zhang et al., 2023; Yang et al., 2023a; Kara et al., 2024; Chen et al., 2023), and canonical representation (Chai et al., 2023; Lee et al., 2023; Ouyang et al., 2024; Kasten et al., 2021). To date, the editing performance of RF-based diffusion transformers has remained largely under-explored. Although (Rout et al., 2024b) employs FLUX (Black-Forest-Labs, 2024) for image editing, its performance is limited to simple tasks such as stylization and face editing while often failing to effectively maintain the structural information of source images. Moreover, currently there is no research exploring the video editing capabilities of RF-based models.

## 3. Method

In this section, we present our method in detail. First, we introduce RF-Solver, which significantly enhances the precision of inversion and reconstruction. Subsequently, we present RF-Edit, an extension of RF-Solver designed to enable high-quality image and video editing.

### 3.1. Preliminaries

Rectified Flow (RF) (Liu et al., 2022b) facilitates the transition between the real data distribution $\pi_0$ and Gaussian Noises distribution $\pi_1$ along a straight path. This is achieved by learning a forward-simulating system defined by the ODE: $d\boldsymbol{Z}_t = v(\boldsymbol{Z}_t, t)dt, t \in [0, 1]$, which maps $\boldsymbol{Z}_1 \in \pi_1$ to $\boldsymbol{Z}_0 \in \pi_0$.

In practice, the velocity field $v$ is parameterized by a neural network $\boldsymbol{v}_\theta$. During training, given empirical observations of two distributions $\boldsymbol{X}_0 \sim \pi_0$, $\boldsymbol{X}_1 \sim \pi_1$ and $t \in [0, 1]$, the forward process (i.e., adding noise) of rectified flow is defined by a simple linear combination: $\boldsymbol{X}_t = t\boldsymbol{X}_1 + (1 - t)\boldsymbol{X}_0$. The differential form of the equation is given by: $d\boldsymbol{X}_t = (\boldsymbol{X}_1 - \boldsymbol{X}_0)dt$. Consequently, the training process optimizes the network by solving the least squares regression problem, which fits the $\boldsymbol{v}_\theta$ with $(\boldsymbol{X}_1 - \boldsymbol{X}_0)$:

$$\min_\theta \int_0^1 \mathbb{E}\left[\|(\boldsymbol{X}_1 - \boldsymbol{X}_0) - \boldsymbol{v}_\theta\left(\boldsymbol{X}_t, t\right)\|^2\right] dt. \quad (1)$$

In the sampling process, the ODE is discretized and solved using the Euler method. Specifically, the rectified flow model starts with a Gaussian noise sample $\boldsymbol{Z}_{t_N} \in \mathcal{N}(0, \boldsymbol{I})$. Given a series of $N$ discrete timesteps $t = \{t_N, ..., t_0\}$, the model iteratively predicts $\boldsymbol{v}_\theta(\boldsymbol{Z}_{t_i}, t_i)$ for $i \in \{N, \cdots, 1\}$ and then takes a step forward until generating the images $\boldsymbol{Z}_{t_0}$, with the following recurrence relation:

$$\boldsymbol{Z}_{t_{i-1}} = \boldsymbol{Z}_{t_i} + (t_{i-1} - t_i)\boldsymbol{v}_\theta(\boldsymbol{Z}_{t_i}, t_i). \quad (2)$$

The RF model can generate high-quality images in much fewer timesteps compared to DDPM (Ho et al., 2020), owing

to the nearly linear transition trajectory established during training. With these effective designs, RF model illustrates great potential in the field of T2I and T2V generation (Black-Forest-Labs, 2024; Zheng et al., 2024).

### 3.2. RF-Solver

The vanilla RF sampler demonstrates strong performance in image and video generation. However, when applied to inversion and reconstruction tasks, we observe significant error accumulation at each timestep. This results in reconstructions that diverge notably from the original image (see Figure 2). This severely limits the performance of RF models in various inversion-based downstream tasks (Hertz et al., 2022; Wang et al., 2024a). Delving into this problem, we identify that the errors stem from the process of estimating the approximate solution for the rectified flow ODE (Wang et al., 2024b; Liu, 2022), which is formulated by Equation (2) in existing methods (Black-Forest-Labs, 2024; Zheng et al., 2024). Consequently, obtaining more precise solutions for the ODE would effectively mitigate these errors, leading to improved performance.

Based on this analysis, we start by carefully examining the differential form of the Rectified flow: $d\boldsymbol{Z}_t = \boldsymbol{v}_\theta(\boldsymbol{Z}_t, t)dt$. This ODE is discretized in the sampling process. Given the initial value $\boldsymbol{Z}_{t_i}$, the ODE can be exactly formulated using the *variant of constant* method:

$$\boldsymbol{Z}_{t_{i-1}} = \boldsymbol{Z}_{t_i} + \int_{t_i}^{t_{i-1}} \boldsymbol{v}_\theta(\boldsymbol{Z}_\tau, \tau)d\tau. \tag{3}$$

In the above formula, $\boldsymbol{v}_\theta(\boldsymbol{Z}_\tau, \tau)$ is the non-linear component parameterized by the complex neural network, which is difficult to approximate directly. As an alternative, we employ the Taylor expansion at $t_i$ to approximate this term:

$$\boldsymbol{v}_\theta(\boldsymbol{Z}_\tau, \tau) = \sum_{k=0}^{n-1} \frac{(\tau - t_i)^k}{k!} \boldsymbol{v}_\theta^{(k)}(\boldsymbol{Z}_{t_i}, t_i) + \mathcal{O}\big((\tau - t_i)^n\big), \tag{4}$$

where $\boldsymbol{v}_\theta^{(k)}(\boldsymbol{Z}_{t_i}, t_i) = \frac{\mathrm{d}^k \boldsymbol{v}_\theta(\boldsymbol{Z}_{t_i}, t_i)}{\mathrm{d}t^k}$, denoting the $k$-order derivative of $\boldsymbol{v}_\theta$ and $\mathcal{O}$ denotes higher-order infinitesimals. Substituting Equation (4) into the integral term yields:

$$\int_{t_i}^{t_{i-1}} \boldsymbol{v}_\theta(\boldsymbol{Z}_\tau, \tau)\, d\tau = \sum_{k=0}^{n-1} \boldsymbol{v}_\theta^{(k)}(\boldsymbol{Z}_{t_i}, t_i) \int_{t_i}^{t_{i-1}} \frac{(\tau - t_i)^k}{k!}\, d\tau + \mathcal{O}\big((\tau - t_i)^n\big). \tag{5}$$

Through the above process, the network prediction term and its higher-order derivatives are separated from the integral. Then we notice that the remaining component in the integral can be computed analytically:

$$\int_{t_i}^{t_{i-1}} \frac{(\tau - t_i)^k}{k!}d\tau = \left[\frac{(\tau - t_i)^{k+1}}{(k+1)!}\right]_{t_i}^{t_{i-1}} = \frac{(t_{i-1} - t_i)^{k+1}}{(k+1)!}. \tag{6}$$

Substituting Equation (6) and Equation (5) into Equation (3), we derive the $n$-th order solution of Rectified flow ODE:

$$\boldsymbol{Z}_{t_{i-1}} = \boldsymbol{Z}_{t_i} + \sum_{k=0}^{n-1} \frac{(t_{i-1} - t_i)^{k+1}}{(k+1)!} \boldsymbol{v}_\theta^{(k)}(\boldsymbol{Z}_{t_i}, t_i) + \mathcal{O}\big(h_i^{n+1}\big), \tag{7}$$

where $h_i := t_{i-1} - t_i$. Equation (7) indicates that to estimate $\boldsymbol{Z}_{t_{i-1}}$, we need to obtain the $k$-th order derivatives $\{\boldsymbol{v}_\theta^{(k)}(\boldsymbol{Z}_{t_i}, t_i)\}$ for $k \in \{0, \cdots, n-1\}$.

Considering $n = 1$, the formula reduces to the standard rectified flow (i.e.,, Equation (2)). In our experiments, we find that setting $n = 2$ effectively mitigates the errors, yielding:

$$\boldsymbol{Z}_{t_{i-1}} = \boldsymbol{Z}_{t_i} + (t_{i-1} - t_i)\boldsymbol{v}_\theta(\boldsymbol{Z}_{t_i}, t_i) + \frac{1}{2}(t_{i-1} - t_i)^2 \boldsymbol{v}_\theta^{(1)}(\boldsymbol{Z}_{t_i}, t_i). \tag{8}$$

Note that $\boldsymbol{v}_\theta^{(1)}$ in Equation (8) is the first-order derivative of the network prediction term $\boldsymbol{v}_\theta$, which cannot be analytically derived due to the complex architecture of the neural network. To estimate this term, we first obtain the network prediction $\hat{\boldsymbol{v}}_{t_i}$ at the timestep $t_i$, i.e., $\hat{\boldsymbol{v}}_{t_i} = \boldsymbol{v}_\theta(\boldsymbol{Z}_{t_i}, t_i)$. Then we step forward a small timestep $\Delta t$ (which is set to $0.01$ in experiments), and update the latents to obtain $\boldsymbol{Z}_{t_i+\Delta t} = \boldsymbol{Z}_{t_i} + \Delta t \cdot \hat{\boldsymbol{v}}_{t_i}$. Subsequently, we calculate an additional prediction of the network at the timestep $t_i + \Delta t$, i.e., $\hat{\boldsymbol{v}}_{t_i+\Delta t} = \boldsymbol{v}_\theta(\boldsymbol{Z}_{t_i+\Delta t}, t_i + \Delta t)$. With $\hat{\boldsymbol{v}}_{t_i}$ and $\hat{\boldsymbol{v}}_{t_i+\Delta t}$, the first-order derivative of $\boldsymbol{v}_\theta$ at the timestep $t_i$ can be estimated as: $\boldsymbol{v}_\theta^{(1)}(\boldsymbol{Z}_{t_i}, t_i) = \frac{\hat{\boldsymbol{v}}_{t_i+\Delta t} - \hat{\boldsymbol{v}}_{t_i}}{\Delta t}$. Substituting this formulation into Equation (8) results in the practical implementation of the RF-Solver algorithm. The complete sampling process for RF-Solver is presented in Algorithm 1.

Obtaining the sampling form of RF-Solver, we further derive its inversion form. Inversion maps data back into noise, which reverses the sampling process. Following previous methods for DDIM inversion (Song et al., 2021a; Dhariwal & Nichol, 2021), the ODE process can be directly reversed in the limit of small steps. Based on this assumption, the inversion process of RF-Solver can be derived as:

$$\widetilde{\boldsymbol{Z}}_{t_{i+1}} = \widetilde{\boldsymbol{Z}}_{t_i} + (t_{i+1} - t_i)\boldsymbol{v}_\theta(\widetilde{\boldsymbol{Z}}_{t_i}, t_i) + \frac{1}{2}(t_{i+1} - t_i)^2 \boldsymbol{v}_\theta^{(1)}(\widetilde{\boldsymbol{Z}}_{t_i}, t_i), \tag{9}$$

where $\widetilde{\boldsymbol{Z}}_{t_i}$ and $\widetilde{\boldsymbol{Z}}_{t_{i+1}}$ denotes the latents during inversion. Through the high order expansion, the error of the ODE solution in each timestep is reduced from $\mathcal{O}\big((h_i)^2\big)$ to $\mathcal{O}\big((h_i)^3\big)$, significantly facilitating inversion and reconstruction process (see Figure 2). Beyond this, RF-Solver can also be applied to any RF-based model (such as FLUX (Black-Forest-Labs, 2024) and OpenSora (Zheng et al., 2024)) for other tasks such as sampling and editing, enhancing performance without requiring additional training.

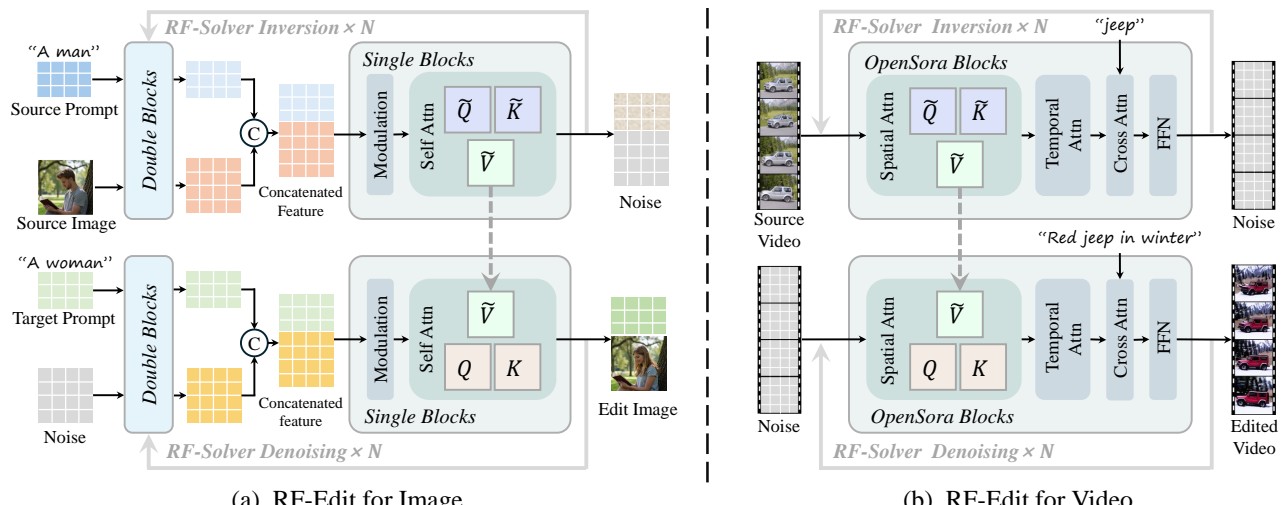

(a). RF-Edit for Image        (b). RF-Edit for Video

*Figure 3.* **RF-Edit pipelines for image editing and video editing.** We design two sub-modules for applying RF-Edit to (a). Image editing with FLUX (Black-Forest-Labs, 2024) and (b). Video editing with OpenSora (Zheng et al., 2024). Note that for FLUX, there are multiple Double Blocks, followed by multiple Single Blocks. For OpenSora, there are multiple OpenSora DiT blocks. For simplicity, only one block of each type is depicted in the figure.

## 3.3. RF-Edit

Incorporating higher-order terms enables RF-Solver to significantly reduce errors in the ODE-solving process, thereby enhancing both sampling quality and inversion accuracy. Furthermore, we extend the application of RF-Solver to the more complex real-world image and video editing tasks, which present greater challenges than reconstruction. In such scenarios, preserving the content and structure of the original image is crucial. For example, when replacing an object in a source image with another one, regions unrelated to the object in this image are expected to remain unaffected by the editing process. However, directly applying RF-Solver during the inversion and denoising stages may cause the model to be overly influenced by the target prompt, resulting in unintended modifications in other regions of the source image or video. Similar issues are common across various existing editing methods (Rout et al., 2024b; Hertz et al., 2022; Tumanyan et al., 2023).

To address this problem, we propose RF-Edit, which builds upon the diffusion transformer architecture. Specifically, we focus on the self-attention layer in the last $M$ transformer blocks of $\boldsymbol{v}_\theta$ at the last $n$ timesteps during inversion. The self-attention operation can be formulated by:

$$\widetilde{\boldsymbol{F}}_{t_k}^m = \text{Attention}(\widetilde{\mathcal{Q}}_{t_k}^m, \widetilde{\mathcal{K}}_{t_k}^m, \widetilde{\mathcal{V}}_{t_k}^m). \tag{10}$$

Here, $k \in \{N-n, \cdots, N\}$, and $m \in \{1, \cdots, M\}$, $\widetilde{\boldsymbol{F}}_{t_k}^m$ denotes the output feature of the self-attention module and $\widetilde{\mathcal{Q}}_{t_k}^m, \widetilde{\mathcal{K}}_{t_k}^m, \widetilde{\mathcal{V}}_{t_k}^m$ represent query, key and value for attention during the inversion process, respectively. We extract and store the Value feature $\{\widetilde{\mathcal{V}}_{t_k}^m\}$ and $\{\widetilde{\mathcal{V}}_{t_k+\Delta t_k}^m\}$ in the process

of RF-Solver algorithm (Algorithm 1):

$$\{\widetilde{\mathcal{V}}_{t_k}^m\} = \text{Extract}\big(\boldsymbol{v}_\theta(\widetilde{\boldsymbol{Z}}_{t_k}, t_k)\big) \tag{11}$$

$$\{\widetilde{\mathcal{V}}_{t_k+\Delta t_k}^m\} = \text{Extract}\big(\boldsymbol{v}_\theta(\widetilde{\boldsymbol{Z}}_{t_k+\Delta t_k}, t_k + \Delta t_k)\big). \tag{12}$$

During the first $n$ timesteps of denoising, considering the $m$th transformer block at the timestep $k$, the original self-attention can be formulated as:

$$\boldsymbol{F}_{t_k}^m = \text{Attention}(\mathcal{Q}_{t_k}^m, \mathcal{K}_{t_k}^m, \mathcal{V}_{t_k}^m), \tag{13}$$

where $\boldsymbol{F}_{t_k}^m$ denotes the output feature of the self-attention module and $\mathcal{Q}_{t_k}^m, \mathcal{K}_{t_k}^m, \mathcal{V}_{t_k}^m$ represent query, key and value for attention during the denoising process, respectively.

In RF-Edit, the above self-attention mechanism is modified to cross-attention where $\mathcal{V}_{t_k}^m$ is replaced by $\widetilde{\mathcal{V}}_{t_k}^m$,

$$\boldsymbol{F}_{t_k}^{m\prime} = \text{Attention}(\mathcal{Q}_{t_k}^m, \mathcal{K}_{t_k}^m, \widetilde{\mathcal{V}}_{t_k}^m). \tag{14}$$

The modified output feature $\boldsymbol{F}_{t_k}^{m\prime}$ is then passed to the subsequent modules for further processing.

Similarly, this feature-sharing process is also adopted in the derivative calculation process of RF-Solver:

$$\boldsymbol{F}_{t_k+\Delta t_k}^{m\prime} = \text{Attention}(\mathcal{Q}_{t_k+\Delta t_k}^m, \mathcal{K}_{k+\Delta t_k}^m, \widetilde{\mathcal{V}}_{k+\Delta t_k}^m). \tag{15}$$

The proposed RF-Edit framework enables high-quality editing while effectively preserving the structural information of the source image/video. Building on this concept, we design two sub-modules for RF-Edit, specifically tailored for image editing and video editing (Figure 3). For image editing, RF-Edit employs FLUX (Black-Forest-Labs, 2024) as the backbone, which comprises several double blocks and single blocks. Double blocks independently modulate

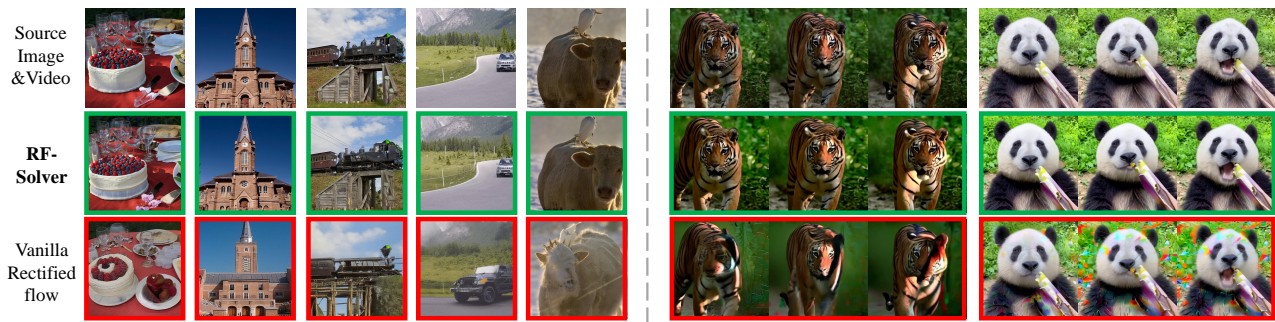

(a). Image Inversion and Reconstruction          (b). Video Inversion and Reconstruction

*Figure 4.* **Qualitative results of image and video reconstruction**. Our method (the second row) demonstrates superior performance compared to the vanilla rectified flow baselines (the third row).

*Table 1.* **Quantitative results on text-to-image generation.** RF-Solver outperforms several baselines.

|  | DPMSolver++ | RF | RF-Heun | **Ours** |
|---|---|---|---|---|
| FID (↓) | 24.63 | 25.33 | 24.40 | **23.98** |
| CLIP Score (↑) | 30.62 | 31.01 | 31.03 | **31.09** |

text and image features, while single blocks concatenate these features for unified modulation. In this architecture, RF-Edit shares features within the single blocks, as they capture information from both the source image and the source prompt, enhancing the ability of the model to preserve the structural information of the source image. For video editing, RF-Edit mainly employs OpenSora (Zheng et al., 2024) as the backbone. The DiT blocks in OpenSora include spatial attention, temporal attention, and text cross-attention. Within this architecture, the structural information of the source video is captured in the spatial attention module, where we implement feature sharing.

## 4. Experiment

### 4.1. Setup

We implement our method respectively on FLUX (Black-Forest-Labs, 2024) and OpenSora (Zheng et al., 2024). In the experiment, we adopt the guidance-distilled variant of FLUX (Black-Forest-Labs, 2024) for image tasks and Open-Sora (Zheng et al., 2024) for video tasks. The derivative computation in RF-Solver requires an additional forward pass, resulting in the network needing to forward twice at each timestep. As a result, when comparing our method with the Rectified Flow baselines, *we set the number of timesteps for the vanilla Rectified Flow to be* **twice** *that of our method* to ensure a fair comparison under the same number of function evaluations (NFE). More detailed information for experiment setup is provided in Appendix B.

### 4.2. Text-to-image Sampling

We compare the performance of our method with DPM-Solver++ (Lu et al., 2022), the vanilla RF sampler, and Heun sampler on the text-to-image generation task. Both the

*Table 2.* **Quantitative results on inversion and reconstruction.** Our method significantly improves the accuracy of reconstruction for both images and videos.

|  | Method | MSE (↓) | LPIPS (↓) | SSIM (↑) | PSNR (↑) |
|---|---|---|---|---|---|
| image | RF | 0.0268 | 0.6253 | 0.7626 | 28.28 |
|  | RF-Heun | 0.0117 | 0.4696 | 0.8924 | 29.67 |
|  | **Ours** | **0.0092** | **0.4239** | **0.9276** | **29.89** |
| video | RF | 0.0206 | 0.4159 | 0.8134 | 18.12 |
|  | RF-Heun | 0.0156 | 0.3554 | 0.8711 | 18.29 |
|  | **Ours** | **0.0134** | **0.3287** | **0.8812** | **18.41** |

quantitative (Section 4.1) and qualitative results (Figure 10) demonstrate the superior performance of RF-Solver in fundamental T2I generation tasks, producing higher-quality images that align more closely with human cognition.

### 4.3. Inversion and Reconstruction

We conduct experiments on inversion and reconstruction for both image and video modalities, comparing our method with the vanilla RF sampler and the Heun sampler.

**Quantitative Comparison**. The quantitative comparisons (Section 4.3) are conducted to reflect the similarity between the source and reconstruction results. Our method demonstrates superior performance across all four metrics compared with the vanilla RF sampler and Heun sampler.

**Qualitative Comparison**. RF-Solver effectively reduces the error in the solution of RF ODE, thereby increasing the accuracy of the reconstruction. As illustrated in Figure 4(a), the image reconstruction results using vanilla rectified flow exhibit noticeable drift from the source image, with significant alterations to the appearance of subjects in the image. For video reconstruction, as shown in Figure 4(b), the baseline reconstruction results suffer from distortion. In contrast, RF-Solver significantly alleviates these issues, achieving more satisfactory results.

### 4.4. Editing

We conduct experiments to evaluate the image and video editing performance of our method. For image editing, we

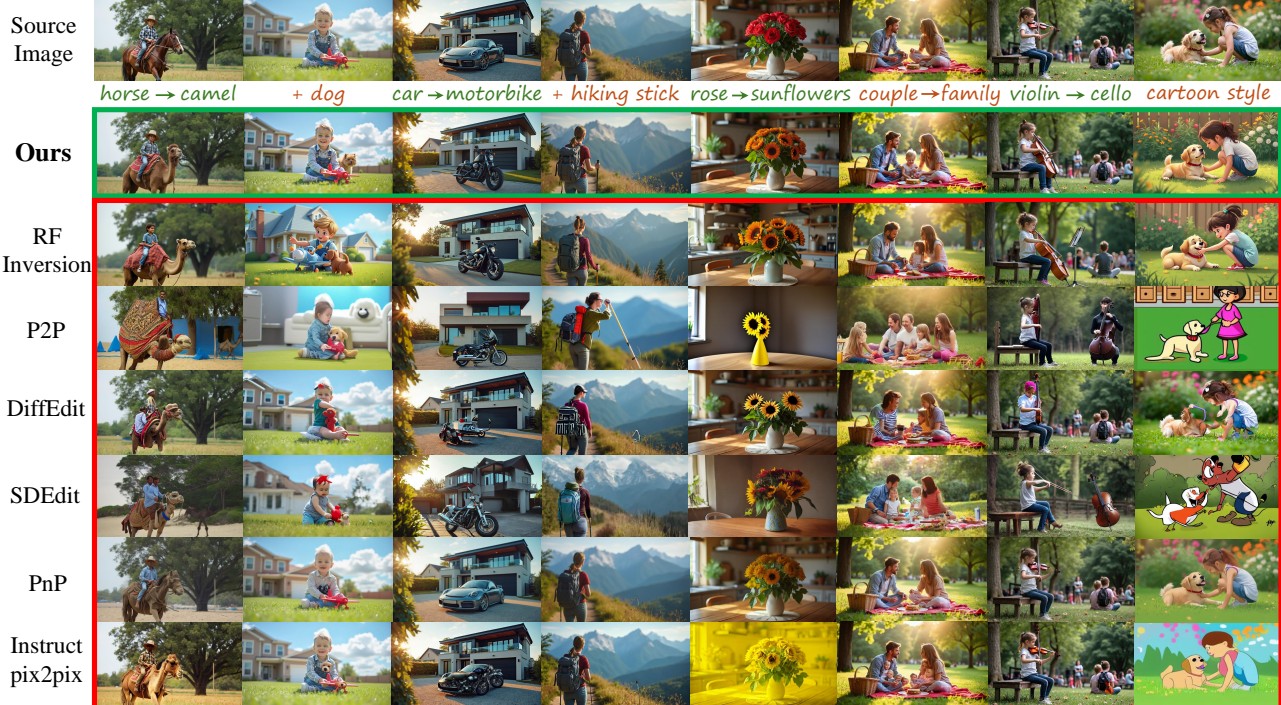

*Figure 5.* **Qualitative comparison of image editing.** With RF-Solver and feature-sharing mechanism in RF-Edit, our method can successfully handle various kinds of image editing cases, outperforming the previous SOTA methods.

*Table 3.* **Quantitative results of image editing.** RF-Edit effectively edits the images according to the prompts while preserving the integrity of unrelated regions.

| | P2P | DiffEdit | SDEdit | PnP | Pix2Pix | RF-Inv | **Ours** |
|---|---|---|---|---|---|---|---|
| LPIPS (↓) | 0.419 | 0.157 | 0.394 | **0.080** | 0.155 | 0.318 | 0.149 |
| CLIP Score (↑) | 30.70 | 32.68 | 31.61 | 30.58 | 32.33 | 33.02 | **33.66** |

*Table 4.* **Quantitative results of video editing.** RF-Edit outperforms several previous SOTA video editing methods.

| | FateZero | Flatten | COVE | RAVE | Tokenflow | **Ours** |
|---|---|---|---|---|---|---|
| SC (↑) | 0.9382 | 0.9420 | 0.9433 | 0.9292 | 0.9439 | **0.9501** |
| MS (↑) | 0.9611 | 0.9528 | 0.9697 | 0.9519 | 0.9632 | **0.9712** |
| AQ (↑) | 0.6092 | 0.6329 | 0.6717 | 0.6586 | 0.6742 | **0.6796** |
| IQ (↑) | 0.6898 | 0.7024 | 0.7163 | 0.6917 | 0.7128 | **0.7207** |

compare our method with P2P (Hertz et al., 2022), DiffEdit (Couairon et al., 2023), SDEdit (Meng et al., 2022), PnP (Tumanyan et al., 2023), Pix2pix (Parmar et al., 2023) and RF-Inversion (Rout et al., 2024b). For video editing tasks, we compare our method with FateZero (Qi et al., 2023), FLATTEN (Cong et al., 2023), COVE (Wang et al., 2024c), RAVE (Kara et al., 2024), Tokenflow (Geyer et al., 2023).

**Quantitative Comparison**. In image editing, Our method outperforms all other methods in CLIP score (Section 4.4), indicating that the edited images align well with the user-provided prompts. For LPIPS, it is noted that PnP (Tumanyan et al., 2023) has a much lower value than all other methods. Based on the qualitative results (Figure 5), it can be seen that PnP is only suitable for editing cases that do not significantly modify the structure or shape of the source image (such as changing red roses into yellow sunflowers). It fails in the case of shape editing, resulting in an image very similar to the source. Consequently, although PnP has the lowest LPIPS score, its CLIP score is the lowest. For video editing, RF-Edit achieves higher scores on VBench (Huang et al., 2024b) metrics (Section 4.4). The results illustrate that our method successfully maintains temporal

consistency while demonstrating superior quality.

**Qualitative Comparison**. For image editing, we compare the performance of our method with several baselines across different types of editing requirements including adding, replacing, and stylization (Figure 5 and Figure 8). The baseline methods often suffer from background changes or fail to perform the desired edits. In contrast, our method demonstrates satisfying performance, effectively achieving a balanced trade-off between the fidelity to the target prompt and the preservation of the source image.

The qualitative results for video editing are shown in Figure 6. RF-Edit illustrates impressive performance in *handling complicated editing* cases (e.g., modifying the leftmost lion among three lions into a white polar bear and changing the other two small lions into orange tiger cubs), whereas all other baseline methods fail in this scenario.

Besides, HunyuanVideo (Kong et al., 2024) has recently demonstrated strong performance in text-to-video generation. Thanks to the generality of our method, it can be implemented on HunyuanVideo for video editing. More

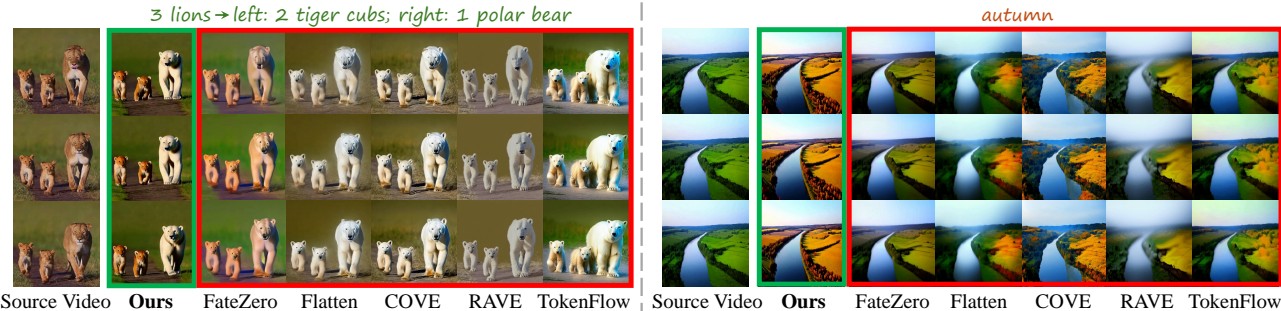

Figure 6. **Qualitative comparison of video editing.** The first video comprises 200 frames with a resolution of $512 \times 512$, while the second video contains 60 frames with a resolution of $1024 \times 768$ (frames are compressed for a neat layout).

qualitative results are shown in Figure 9.

### 4.5. Ablation Study

We conduct ablation studies to illustrate the effectiveness of RF-Solver and RF-Edit. Without loss of generality, these ablation studies are performed on the image tasks using FLUX (Black-Forest-Labs, 2024) as the base model.

**Taylor Expansion Order of RF-Solver.** We investigated the impact of the Taylor expansion order in RF-Solver (Section 4.5) under the same NFE across different orders. Compared to the first-order expansion (i.e., the vanilla rectified flow), the second-order expansion demonstrates a significant improvement across various tasks. However, higher-order expansions do not yield further enhancements. We speculate that this is primarily due to higher-order Taylor expansions requiring more inference steps per timestep. With a fixed NFE, this results in a reduced overall number of timesteps compared to lower-order expansions, leading to suboptimal performance. Moreover, computing the higher-order derivatives of $v_\theta(Z_{t_i}, t_i)$ substantially increases the complexity of the algorithm, posing challenges for practical applications. Consequently, we employ second-order expansion (i.e., RF-Solver-2 in Section 4.5) for various downstream tasks due to its effectiveness and simplicity.

Table 5. **Ablation study on the Taylor Expansion order.** Here, RF implies the *vanilla Rectified Flow* without the proposed RF-Solver algorithm. All of the experiments for editing in the table apply the proposed feature-sharing mechanism for better results.

|  | Metric | RF | **RF-Solver-2** | RF-Solver-3 |
|---|---|---|---|---|
| Sampling | FID ($\downarrow$) | 25.33 | 23.98 | **23.94** |
|  | CLIP Score ($\uparrow$) | 31.01 | **31.09** | 31.09 |
| Inversion | MSE ($\downarrow$) | 0.0268 | **0.0092** | 0.0131 |
|  | LPIPS ($\downarrow$) | 0.6253 | **0.4239** | 0.4817 |
| Editing | LPIPS ($\downarrow$) | 0.1524 | **0.1494** | 0.1503 |
|  | CLIP Score ($\uparrow$) | 32.97 | **33.66** | 33.18 |

**Feature Sharing Steps of RF-Edit.** RF-Edit leverages feature sharing to maintain the structural consistency between original images and edited images. However, an excessive number of feature-sharing steps may result in the edited

output being overly similar to the source image, ultimately undermining the intended editing objectives (Figure 7). To investigate the impact of feature-sharing steps on editing results, we incrementally increase the number of feature-sharing steps applied to the same image. Due to the varying levels of difficulty that different images presented to the model, the optimal number of sharing steps may differ across cases. Experimental results reveal that setting the sharing step to 5 effectively meets the editing requirements for most images. Additionally, we can customize the sharing step for each image to identify the most satisfying outcome.

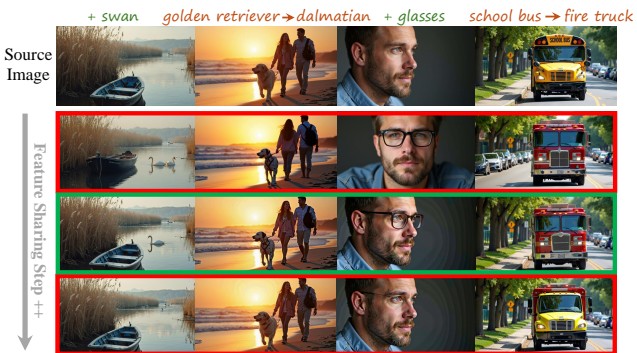

Figure 7. **Ablation study of feature-sharing step in RF-Edit.** The second row is the results produced *solely by RF-Solver*, without the proposed feature-sharing mechanism. Feature-sharing can significantly enhance the consistency between source and target images, while a too-large feature-sharing step may lead to the failure of editing.

### 5. Conclusion

In this paper, we propose RF-Solver, a versatile sampler for the rectified flow model that solves the rectified flow ODE with reduced error, thus enhancing the image and video generation quality across various tasks such as sampling and reconstruction. Based on RF-Solver, we further propose RF-Edit, which achieves high-quality editing performance while effectively preserving the structural information in source images or videos. Extensive experiments demonstrate the versatility and effectiveness of our method.

## Acknowledgments

This work was partly supported by Shenzhen Key Laboratory of next generation interactive media innovative technology (No:ZDSYS20210623092001004) and National Key Laboratory of Human-Machine Hybrid Augmented Intelligence, Xi'an Jiaotong University (No. HMHAI202410).

## Impact Statement

This paper presents work whose goal is to advance the field of Machine Learning. There are many potential societal consequences of our work, none which we feel must be specifically highlighted here.

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

# Appendix

# A. Pesudo Code of RF-Solver Algorithm.

---

**Algorithm 1** Sampling process of RF-Solver

---

**Input:**

$\boldsymbol{v}_\theta$            ▷ *Velocity function*

$t = [t_N, \ldots, t_0]$        ▷ *Time steps*

$\boldsymbol{Z}_{t_N} \sim \mathcal{N}(0, I)$      ▷ *Initial Gaussian Noise*

**For** $i = N$ **to** $1$ **do**

   $\hat{\boldsymbol{v}}_{t_i} \leftarrow \boldsymbol{v}_\theta(\boldsymbol{Z}_{t_i}, t_i)$

   $\boldsymbol{Z}_{t_i + \Delta t_i} \leftarrow \boldsymbol{Z}_{t_i} + \Delta t_i \hat{\boldsymbol{v}}_{t_i}$

   $\hat{\boldsymbol{v}}_{t_i + \Delta t_i} \leftarrow \boldsymbol{v}_\theta(\boldsymbol{Z}_{t_i + \Delta t_i}, t_i + \Delta t_i)$

   $\boldsymbol{v}_{t_i}^{(1)} \leftarrow (\hat{\boldsymbol{v}}_{t_i + \Delta t_i} - \hat{\boldsymbol{v}}_{t_i})/\Delta t_i$    ▷ *Calculating the Derivatives*

   $\boldsymbol{Z}_{t_{i-1}} \leftarrow \boldsymbol{Z}_{t_i} + (t_{i-1} - t_i)\hat{\boldsymbol{v}}_{t_i} + \frac{1}{2}(t_{i-1} - t_i)^2 \boldsymbol{v}_{t_i}^{(1)}$

**Output:** $\boldsymbol{Z}_0$

---

# B. Experimental Settings

## B.1. Baselines and Implementation Details

**Text-to-Image Generation.** We compare our methods with the following baselines: FLUX with the vanilla sampler, Heun Solver, and DPM-Solver. The Heun Solver is a second-order ODE solver that can be applied to pretrained rectified flow to solve the ODE more precisely. DPM-Solver is a high-order sampler for diffusion ODE, which is not suitable for RF-based models like FLUX. As an alternative, we apply the DPM-Solver on Stable Diffusion to evaluate its performance. For FLUX with the vanilla sampler and the Heun Solver, we randomly select 10000 images from the MS-COCO validation dataset and use their caption as the prompt for generation. The resolution of generated images is $1024 \times 1024$. For DPM-Solver, we adopt the implementation from the diffuser, adopting its default setting to generate images. The total NFE for generating one image is set to 10 for both our method and baselines.

**Inversion.** We compare the performance of our methods among RF with the vanilla sampler and the Heun sampler. For image inversion, similar to Text-to-Image generation, we use images and captions from the MS-COCO validation set. For video inversion, we select about 40 videos from social media platforms such as TikTok and other publicly available sources. We have observed the quality of the text prompts significantly influences the quality of inversion. Consequently, we employ GPT-4o to generate detailed captions for both images and videos, which are then used in the inversion tasks. The total NFE for generating one image/video is set to 50 for both our method and baselines.

**Editing.** For image editing, we share the features of the last 19 single blocks in FLUX. For video editing, we share the features of the last 14 blocks in Open-Sora. We adjust the hyper-parameter of feature-sharing steps to achieve better results for both image and video editing. For image edit-

ing, we use over 300 images for quantitative comparison, which both include real images (obtained from public social media and the DIV2K dataset) and generated images. For each image, we use GPT-4o to generate the source prompt and manually refine the generated prompt. There are 2 ~3 target prompts for different requirements of editing including adding, replacing, and stylization for each source image. We compare our methods with RF-inversion and several diffusion-based editing methods. For RF-inversion, we adopt the implementation in ComfyUI (com). For other baselines, we use their implementation from *diffuser* and adjust the relevant hyper-parameters to achieve optimal results. For video editing, the data preparation mainly follows previous works COVE (Wang et al., 2024c). We use the official codes of all the baseline methods and tune the hyper-parameters to achieve satisfactory results.

## B.2. Evaluation Metrics

For text-to-image sampling, we report Fréchet Inception Distance (FID) and CLIP Scores. The FID is a metric used to evaluate the quality of generated images by assessing the similarity between the distributions of real and generated image features, typically extracted using a pre-trained Inception network. The CLIP Score evaluates the alignment between generated images and textual descriptions by measuring the similarity of their embeddings within a shared multimodal space using the CLIP model.

For Inversion tasks, our evaluation metrics include MSE, LPIPS, SSIM, and PSNR. MSE measures the average squared difference between predicted and ground-truth values, quantifying the overall error in pixel intensity. LPIPS assesses perceptual similarity between images by comparing deep feature representations extracted from neural networks, aligning with human perception. SSIM evaluates image quality by comparing luminance, contrast, and structure to measure the similarity between the reference and reconstructed images. PSNR quantifies the ratio between the maximum possible signal value and the power of noise, commonly used to assess image reconstruction quality.

For video editing, we adopt the VBench Metrics. The evaluation criteria include Subject Consistency, Motion Smoothness, Aesthetic Quality, and Imaging Quality. Subject Consistency measures whether the subject (e.g., a person) remains consistent throughout the video by computing the similarity of DINO features (Caron et al., 2021) across frames, which is similar to the CLIP Score for images. Motion Smoothness assesses the smoothness of motion in the generated video using motion priors from the video frame interpolation model (Li et al., 2023). Aesthetic Quality evaluates the artistic and visual appeal of each frame as perceived by humans, leveraging the LAION aesthetic predictor (LAION-AI, 2022). Imaging Quality examines the

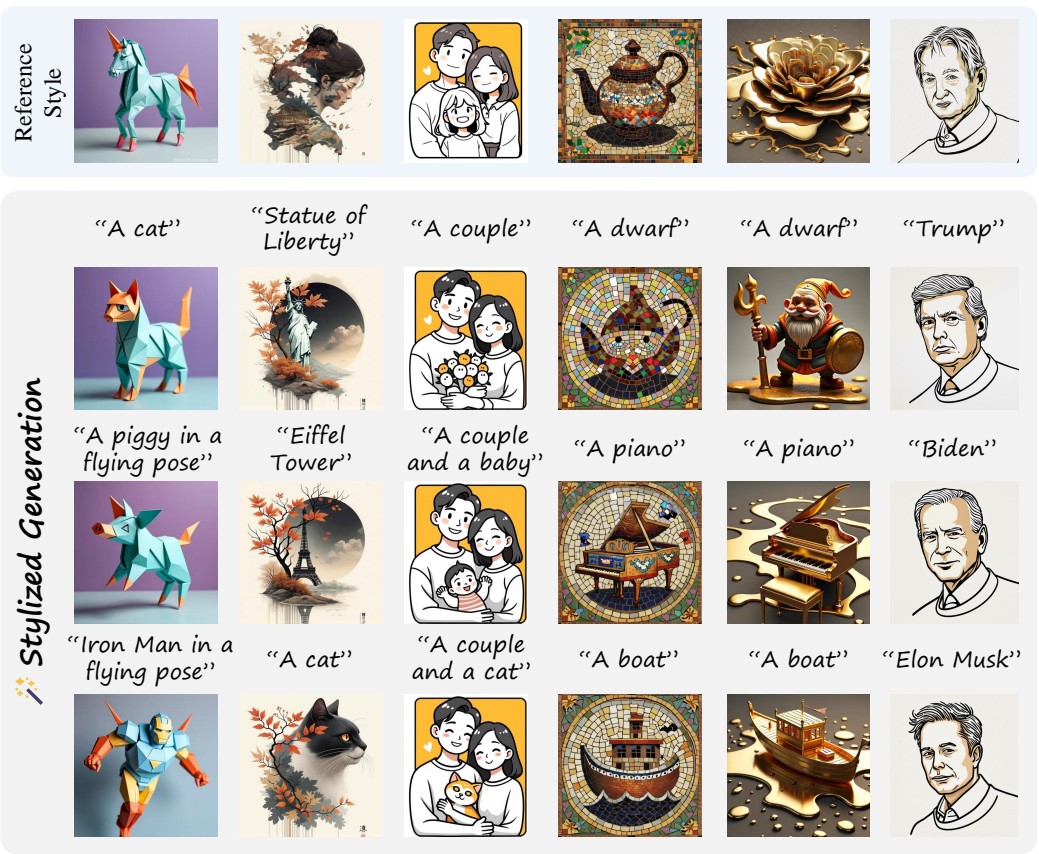

*Figure 8.* **Stylized Generation.**

level of distortion in the generated frames (e.g., blurring or flickering) based on the MUSIQ image quality predictor (Ke et al., 2021).

## C. Qualitative Results for Text-to-image Sampling

Qualitative results for Text-to-image Sampling are shown in Figure 10. Experimental results demonstrate that compared to vanilla Rectified Flow, the RF-Solver sampler can generate higher-quality images that better align with human perception.

## D. More Qualitative Results for Image Editing

Here we provide more qualitative results for image editing and stylization (Figure 8).

## E. More Qualitative Results for Video Editing

HunyuanVideo (Kong et al., 2024) has recently demonstrated strong performance in text-to-video generation. The backbone of HunyuanVideo is similar to FLUX, which also contains several double-stream blocks, followed by single-stream blocks. We implement the RF-Solver and RF-Edit on HunyuanVideo, where the RF-Edit shares the feature in the single-stream block. The results are shown in Figure 9.

## F. More Potential Applications

RF-Solver is a universal sampler for rectified flow. Besides image and video editing, it is also potential on image or video generation (Xiao et al., 2025; Ma et al., 2024b;a; Lin et al., 2025) and other diffusion-based tasks (He et al., 2024; Fang et al., 2024; Guo et al., 2022). Furthermore, our proposed feature-sharing method in RF-Edit can also be applied in other image and video editing methods (Zhu et al., 2024b;a).

Source Video

"jeep"->"pink Porsche"

"parrot"->"dragon"

Edited Video

Source Video

+ "Santa hat"

"rabbit"->"Tom cat"

Edited Video

Source Video

"rabbit"->"cat"

"heart "->"yellow car"

Edited Video

*Figure 9.* **More video editing results.**

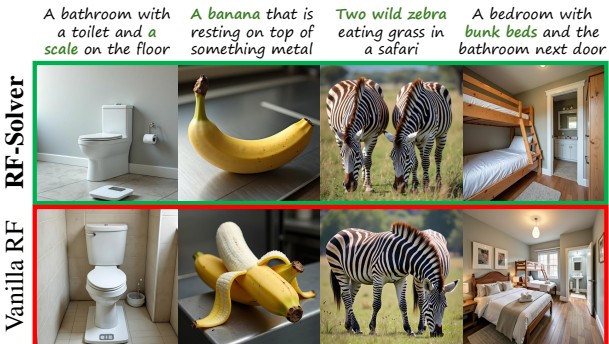

*Figure 10.* **Qualitative results of text-to-image generation.** By employing the RF-Solver, the model is able to generate images with higher quality than baselines.

