# OpenReview forum: "Taming Rectified Flow for Inversion and Editing"
_ICML.cc/2025/Conference — ICML 2025 poster_

### Official Review · Reviewer_5wXG · 2025-02-18

**Overall Recommendation:** 4

**Summary:**

The paper points out the problem of in accurate reconstruction for regular inversion method on flow models, and proposed to methods for inversion and editing. For Inversion, the authors introduced a new ODE by employing higher order with Taylor expansion, and found the second-order expansion is already helpful for better inversion. Though intriducing more computation when approximating the second order term, the authors cut half of the NFE for a fair comparision. The second contributions is to manipulate the self attention layers during the reverse process. The authors applied editing for both image and videos to demonstrtate its effeciveness.

## update after rebuttal
I increased score to accept since the authors addressed my concern on the ablation studies.

**Claims And Evidence:**

Mostly yes, please see experiment section below for more questions.

**Essential References Not Discussed:**

no

**Experimental Designs Or Analyses:**

1. One important contribution of the paper is to use higher order RF ODE for better inversion/reconstruction. I would like to see how much of improvement it brings up. Though Table 5 compares under the same NFE, it does not show the advantage of the proposed higher-order ODE as it increases. I understand that RF-Solver-2 might be a good sweet spot for tradeoff performance and latency, but there’s no experiment to demonstrate the effectiveness when increasing the order using Tyler explosion, which is the key idea of the paper. Specifically, I’m mainly looking for the proof of the intuition behind the methods, e.g., you find higher order ODE brings better preformance but more computations expensive, and then you figure out using less NFE to resolve it and fair comparison, or, on the other hand, increase the steps used in regular inversion/editing with same NFE for fair comparison.

2. An important problem in many inversion-based editing methods are the tuning of hyperparameters. It's mentioned in the last section that  sharing step 5 is a good choice, do you have more studies on it? e.g., when to apply to early stage of later, what's the hyperparameter robustness for different editing (e.g. stylization, adding object, changing object)?

3. what happen if you only have RF-solver but not RF-edit for editing? this mean you only have RF-solver to better preserve the original construction of the image and use prompt for editing without changing the attention modules. This would better demonstrate the advantage of RF-edit.

4. What's the memory for storing Vs? Why not storing/sharing Ks or attention maps?

**Methods And Evaluation Criteria:**

yes

**Other Comments Or Suggestions:**

refer to the other secions

**Other Strengths And Weaknesses:**

refer to the other secions

**Questions For Authors:**

refer to the other secions

**Relation To Broader Scientific Literature:**

The method proposed in this paper differs from other works by emplying taylor expansion to construct a higher order ODE for flow model inversion. The editing part is good but less significant because attention-based editing has been explored a lot in previous works.

**Theoretical Claims:**

The equations in the paper look correct to me.

---

> ### Author Rebuttal · Authors · 2025-04-01
>
> Dear Reviewer 5wXG,
>
> Thank you for your comprehensive and detailed review of our paper and the recognition of our work's effectiveness. We provide our feedback as follows.
>
> > Experiment to demonstrate the effectiveness when increasing the order using Taylor expansion, which is the key idea of the paper.
>
> Thanks for your advice! We have provided experiments on both image generation and reconstruction in [Table5](https://postimg.cc/4mxRLZNc).
>
> The results illustrate that under various timesteps, our methods outperform the baseline significantly. With the higher order expansion (RF-Solver-3), the performance becomes even better. Although the comparison is conducted with the same number of timesteps (rather than the same NFE), we notice that under similar NFE, a higher order expansion sometimes also illustrates a better performance. For example, considering the 20 timesteps for Vanilla RF, 10 timesteps for RF-Solver-2, and 7 timesteps for RF-Solver-3, the NFE for them is similar (20, 20, 21, respectively), while RF-Solver-3 illustrates the best performance among them.
>
> > Hyperparameter robustness for feature sharing
>
> We have provided a more detailed analysis about the choice of feature sharing in [Figure11](https://postimg.cc/p5QJNgW5), [Figure12](https://postimg.cc/9rVBPFbd), [Figure13](https://postimg.cc/CdbNffGM), [Figure14](https://postimg.cc/F7ZVT5C2). In our work, solely tuning the hyperparameter of the feature-sharing step is enough to obtain a satisfying result. What's more, editing a high-resolution image (1360 \* 768) using our methods only takes a short time (less than 1 minute). As a result, we believe that the parameter-tuning is acceptable for most users.
>
> > What happens if you only have RF-solver but not RF-edit for editing
>
> As mentioned in Line 242~258 in the main paper, only using RF-Solver for editing sometimes cannot maintain the consistency between the source image and the target image. Figure 7 illustrates some results produced solely by RF-Solver. We also provide a quantitative ablation study about this in [Table7](https://postimg.cc/dky96hgB).
>
>
> > What's the memory for storing Vs? Why not storing/sharing Ks or attention maps?
>
> For editing a 1360\*768 image, the total memory needed to store the feature is about 18G. We store the feature in the CPU, rather than GPU Memory. In practice, this would not significantly reduce the efficiency of inversion and editing.
>
> We also add the qualitative results about sharing K or attention maps in [Figure13](https://postimg.cc/CdbNffGM). Experimental results demonstrate that, compared to K or the attention map, V contains the richest information regarding the original image. Sharing V can effectively preserve the source image's details in a relatively small number of feature-sharing steps, whereas sharing K and the attention map under the same number of steps yields less satisfactory outcomes. Therefore, we choose to share V for its effectiveness and efficiency.

---

> > ### Comment · Reviewer_5wXG · 2025-04-06
> >
> > I appreciate the authors for the responses and additional results. My concerns are addressed and I will increase my rating to accept. The additional ablation studies explains the design choice and the effectiveness of the proposed methods.

---

### Official Review · Reviewer_1xw9 · 2025-03-12

**Overall Recommendation:** 3

**Summary:**

This paper proposes a flow inversion method based on the improved higher order ODE, which simply extracts the first order derivative from the volatility prediction for more accurate model denoising generation. this method can be applied to accurate image or video inversion based on the pre-trained image or video flow diffusion models. To keep more consistency, the author also proposes the reuse the "Value" from the attention for image or video editing, which results in better consistency on text-guided editing.

**Claims And Evidence:**

yes

**Essential References Not Discussed:**

Most of the references are discussed.

**Experimental Designs Or Analyses:**

The experimental comparison is fair and sufficient.

**Methods And Evaluation Criteria:**

yes, it makes sense to me

**Other Comments Or Suggestions:**

Nice work, I saw there is a hunyuan video editing code; is there any results on this?

**Other Strengths And Weaknesses:**

Strengths: This paper proposes an effective flow diffusion inversion method, and it is training-free, which achieves pretty accurate image and video inversion. Based on the proposed method, we can have more consistent content editing.

Weakness: No significant weakness found.

**Questions For Authors:**

Please refer to above.

**Relation To Broader Scientific Literature:**

DDIM inversion would be the most related literature.

**Theoretical Claims:**

yes

---

> ### Author Rebuttal · Authors · 2025-04-01
>
> Dear Reviewer 1xw9,
>
> We sincerely thank for your recognition of the methods and experiments in our work!
>
> > I saw there is a hunyuan video editing code; is there any results on this?
>
> We provided some video editing results produced by HunyuanVideo in Figure 9 in the Appendix. We also provide the code for video editing in the supplementary.
>
> The quantitative results are shown in the table below, where the experimental settings follow Table 4 in the main paper. We will also add this to the main paper.
>
> **Table 9. Quantitative results about HunyuanVideo**
> | |SC|MS|AQ|IQ|
> |-|-|-|-|-|
> |RF-Edit (HunyuanVideo)   | **0.9573** | **0.9749** | **0.6880**| **0.7298** |
> |RF-Edit (OpenSora)   | 0.9501 | 0.9712| 0.6796 | 0.7207 |

---

> > ### Comment · Reviewer_1xw9 · 2025-04-04
> >
> > Thanks for the great work!

---

> > > ### Author Response · Authors · 2025-04-07
> > >
> > > Dear Reviewer 1xw9,
> > >
> > > We sincerely appreciate your recognition of our work! May we kindly ask whether our response has addressed your concern, and would you consider further increasing the ratings accordingly? Thanks once again for your support!
> > >
> > > Authors

---

### Official Review · Reviewer_eQrG · 2025-03-14

**Overall Recommendation:** 4

**Summary:**

This paper aims to leverage Rectified-flow-based generative models for unified image and vide editing. Specifically, this paper proposes RF-Solver, which uses high-order Taylor expansion to eliminate the errors in the inversion and reconstruction process. The paper further proposes RF-Edit, which use value feature-sharing for editing. Extensive experiments illustrate the effectiveness of the proposed methods.

**Claims And Evidence:**

1.	The paper primarily contributes in two aspects: the RF-Solver sampler and the RF-Edit framework. The RF-Solver sampler is novel and effective. The RF-Edit framework is somewhat similar to existing U-Net-based methods, while this paper explores its application on the mainstream Diffusion Transformer architecture.

2.	The research area of this paper is very active and interesting. Given the performance of rectified-flow-based models in image and video generation, exploring their performance on various downstream tasks is meaningful and promising.

3.	The paper is well-written and easy to follow. The authors provide a theoretical derivation of the high-order expansion for RF-Solver.

**Essential References Not Discussed:**

No

**Experimental Designs Or Analyses:**

Overall, the experiment is extensive and thorough.

**Methods And Evaluation Criteria:**

1. The paper proposes using a high-order Taylor expansion to reduce errors during the inversion and reconstruction processes. Then, it proposes the RF-Edit framework, which uses a feature-sharing mechanism to further preserve unintended modifications, i.e., the background.

2. Extensive experiments from both qualitative and quantitative perspectives illustrate the effectiveness of the methods. The authors explore performance across various backbones, including FLUX and OpenSora, demonstrating the universality of the proposed methods.

**Other Comments Or Suggestions:**

No

**Other Strengths And Weaknesses:**

The paper exhibits several strengths, introducing novel aspects and presented in a clear and well-structured manner. My main concerns are listed below:

1. Some implementation details are unclear, particularly regarding experiments conducted on HunyuanVideo. The authors are expected to provide a more thorough specification of the implementation.

2. The authors should provide qualitative and quantitative results to demonstrate the benefits of RF-Solver without feature-sharing. Additionally, to further validate the superiority of RF-Edit, the authors could include qualitative comparisons between RF-Edit and previous methods such as MasaCtrl and PnP, implemented on the U-Net architecture.

3.  The ablation study in Table 5 maintains the same total number of function evaluations (NFE) across different orders, leading to suboptimal performance for higher-order expansions (e.g., the row labeled 'RF-Solver-3'). The authors should conduct additional ablation studies using the same number of timesteps (instead of total NFE) to better illustrate whether higher-order expansions can further enhance performance.

**Questions For Authors:**

See "Other Strengths and Weaknesses"

**Relation To Broader Scientific Literature:**

The paper aims to achieve satisfying editing outcomes using the recent rectified-flow based DiT. There are extensive works about image and video editing based on diffusion and UNet architecture such as PnP [1] and Masactrl [2]. More recently, there are also some works using FLUX to achieve image editing [3].

[1]. Plug-and-Play Diffusion Features for Text-Driven Image-to-Image Translation

[2]. MasaCtrl: Tuning-free Mutual Self-Attention Control for Consistent Image Synthesis and Editing

[3]. Semantic Image Inversion and Editing using Rectified Stochastic Differential Equations

**Theoretical Claims:**

The theoretical derivation provided in this paper is comprehensive, offering theoretical justification for both sampling and inversion processes.

---

> ### Author Rebuttal · Authors · 2025-04-01
>
> Dear Reviewer eQrG,
>
> Thanks for your comprehensive review and insightful comments on our paper. We appreciate that you recognize the motivation and performance of our methods. The response to your concerns is shown below.
>
> > Implementation Details about HunyuanVideo
>
> Thanks for your advice! For video editing on HunyuanVideo, total timesteps are set to 25, and the feature-sharing step is between 1 to 5 for different cases. We will add these implementation details in the main paper. What's more, we have provided the code in the supplementary materials and planned to release it in the future.
>
> > The authors should provide qualitative and quantitative results to demonstrate the benefits of RF-Solver without feature-sharing.
>
> Thanks for your advice, we have provided the qualitative results of feature-sharing in RF-Edit in [Table7](https://postimg.cc/dky96hgB). The quantitative results are provided in Figure 7 in the paper.
>
> What's more, we also provide a more detailed analysis about the feature sharing strategy in [Figure11](https://postimg.cc/p5QJNgW5), [Figure12](https://postimg.cc/9rVBPFbd), [Figure13](https://postimg.cc/CdbNffGM), [Figure14](https://postimg.cc/F7ZVT5C2). We finally choose to share the V feature in the last 20 single-stream blocks of FLUX in the timesteps near to noise, which is proved to be the most effective choice.
>
> > Qualitative comparisons between RF-Edit and previous methods such as MasaCtrl and PnP.
>
> Thanks for your advice! The results are provided in [Table6](https://postimg.cc/XB6kzVb0) and [Figure15](https://postimg.cc/5X2qG7Zj). The qualitative comparisons between our method and PnP are shown in Figure 5. Our method outperforms the baselines both qualitatively and quantitatively.
>
> > Additional ablation studies using the same number of timesteps
>
> Thanks for your advice. We have provided more detailed experiments on both image generation tasks and reconstruction tasks in [Table5](https://postimg.cc/4mxRLZNc).
>
> The results illustrate that with the increase of Taylor Expansion order, our method illustrates a better performance at various timesteps. Although the comparison is conducted with the same number of timesteps (rather than the same NFE), we notice that under similar NFE, a higher order expansion sometimes also illustrates a better performance. For example, considering the 20 timesteps for Vanilla RF, 10 timesteps for RF-Solver-2, and 7 timesteps for RF-Solver-3, the NFE for them is similar (20, 20, 21, respectively), while RF-Solver-3 illustrates the best performance among them.

---

> > ### Comment · Reviewer_eQrG · 2025-04-08
> >
> > Thank you for your comprehensive responses; I would like to check the video editing results to verify the editing consistency. Could you please provide the video results in the Figure 9?

---

> > > ### Author Response · Authors · 2025-04-08
> > >
> > > Dear Reviewer eQrG,
> > >
> > > Thanks for your reply! Some results are provided as follows:
> > >
> > > | Source Video | Prompt |Edited Video|
> > > |-|-|-|
> > > |[link](https://postimg.cc/SJdDbmrv) | rabbit -> cat | [link](https://postimg.cc/k6wv3cNt) |
> > > |[link](https://postimg.cc/62QGRNRy) | parrot -> dragon| [link](https://postimg.cc/3kBRZc3p) |
> > > |[link](https://postimg.cc/hzVN9HBR)| kangroo-> Tom Cat| [link](https://postimg.cc/XB8hPdq2) |
> > > |[link](https://postimg.cc/pyRPsB1J)| human -> Cat| [link](https://postimg.cc/9rfChQ24) |
> > > |[link](https://postimg.cc/565NxRYG) | + Crown | [link](https://postimg.cc/ctRZLvHY) |
> > > |[link](https://postimg.cc/FfY4JWzh)| heart -> car | [link](https://postimg.cc/WhdhTn6c) |

---

### Official Review · Reviewer_n1SC · 2025-03-18

**Overall Recommendation:** 2

**Summary:**

This paper introduces RF-Solver, a training-free, high-order solver to improve inversion and reconstruction in rectified flow models, and RF-Edit, a feature-sharing mechanism for image and video editing. The method states improvements in inversion accuracy and editing quality compared to vanilla rectified flow and several baselines.

**Claims And Evidence:**

The paper claims that RF-Solver improves inversion accuracy by reducing the error in solving the rectified flow ODE through a high-order Taylor expansion, and that RF-Edit enhances image and video editing by transferring self-attention value features from the inversion process to the denoising process. The authors support these claims with quantitative results (e.g., lower MSE, LPIPS, and higher CLIP scores in the inversion and editing tasks) and qualitative comparisons against several baselines.

However, some details affecting the strength of these claims remain unclear. For instance, the evaluation lacks complete descriptions of dataset selection, the exact number of solver steps used by baseline methods, and a detailed step-vs-quality analysis. In addition, the editing evaluation relies primarily on LPIPS, which may not fully capture semantic consistency; alternative metrics such as CLIP-I or DINO could offer more insight. Finally, the derivation presented in Eq (9) appears equivalent to Heun's method, and it is not fully clear how the proposed method differs from standard second-order approaches.

**Essential References Not Discussed:**

Some very related papers are cited but not discussed:

[1] Xie et al, SANA: EFFICIENT HIGH-RESOLUTION IMAGE SYNTHESIS WITH LINEAR DIFFUSION TRANSFORMERS (2025)
[2] DragDiffusion: Harnessing Diffusion Models for Interactive Point-based Image Editing
[3] Can et al., MasaCtrl: Tuning-Free Mutual Self-Attention Control for Consistent Image Synthesis and Editing
[4] Chung et al, Style Injection in Diffusion: A Training-free Approach for Adapting Large-scale Diffusion Models for Style Transfer

**Experimental Designs Or Analyses:**

The authors valuate image editing with LPIPS, but that metric often fails to measure fine-grained editing fidelity. They do not include widely used metrics like CLIP-I or DINO, which may capture semantic changes more precisely.

The effect of solver order 2 vs 3 is briefly mentioned, yet the explanation for 3rd order being worse remains superficial (“less timesteps overall”). More thorough ablations or error analyses would help.

**Methods And Evaluation Criteria:**

The evaluation method and clarity could be improved. The paper does not explain how the evaluation data were chosen or how large each set was. The authors also do not describe in detail how many solver steps are used by the baseline models. They claim a faster solver but do not show a thorough step-vs-quality analysis. Feature-sharing for editing is described, but it is similar to ideas in works like DragDiffusion or MasaCtrl. The paper does not include direct comparisons or discussions that clarify how this method stands out.

Table 3 depends strongly on LPIPS, known to be uninformative in some editing tasks. Authors do not attempt alternative metrics (CLIP-I, DINO, or MDINO) that better capture semantic consistency.

Doubling baseline solver steps to match function evaluations is not necessarily fair since most baseline methods achieve pareto-optimality  in lower number of steps.

**Other Comments Or Suggestions:**

- Please confirm if eq(9) is indeed identical to Heun. If there is novelty, show a direct comparative formula.
- Provide more data on number of steps vs. final image quality.
- Include key experimental details (datasets, NFE, hyperparameters) in the main text to improve reproducibility.
- Consider more robust metrics (e.g., CLIP-I, DINO) for evaluating editing quality to complement LPIPS.
- Clarify and justify choices in feature-sharing (number of layers and timesteps) with further ablations or discussion.
- Discuss potential failure cases or limitations, especially regarding inversion prompt dependency and computational cost.
- Show comprehensive experiments against stonger baslines (i.e. SANA) to highlight improvement.

**Other Strengths And Weaknesses:**

N/A

**Questions For Authors:**

**I am adding here extra questions for the authors because there is an issue with the visibility of my comments.**


## New questions April 8th

I thank the authors for their response. The extra experiments are welcome.

In light of your response, the method you propose is part of the well known family of Taylor series integrators. To my understanding this type of integrator **have** been extensively studied for diffusion models in the probability flow ODE setting [1-3] . Could provide further clarifications about the novelty of your method and how it differs from previous work?

In Table 5 does the number of steps indicate the same number of NFE?

[1] [DEIS](https://arxiv.org/abs/2204.13902)

[2] [DPM-Solver](https://arxiv.org/abs/2211.01095)

[3] [DPM-Solver++](https://arxiv.org/pdf/2206.00927)

**Relation To Broader Scientific Literature:**

They cite SANA 2025 for diffusion-based high-resolution image synthesis, and mention DragDiffusion, MasaCtrl, and Style Injection for attention-based editing. But the paper itself does not seriously compare with those or situate how its approach truly diverges from known feature-sharing or other inversion techniques.

**Theoretical Claims:**

The derivations (e.g., on Taylor expansion) are technically solid. Same for the attention sharing.

But if I understand it correctly your step update in Eq(9) is equivalent to Heun's method:
$
Z_{t+\Delta t} = Z_t + \Delta t u_t + \frac{1}{2} (\Delta t)^2 \frac{u_{t+\Delta t}- u_t}{\Delta t}
= Z_t +  \Delta t u_t + \frac{1}{2} \Delta t (u_{t+\Delta t}- u_t)
= Z_t +  \frac{\Delta t }{2} \Delta t (u_{t+\Delta t}+ u_t)
$
Could you please explain how your method differs from other 2nd order methods?

---

> ### Author Rebuttal · Authors · 2025-04-01
>
> Dear Reviewer n1SC,
>
> Thanks for your time and thoughtful review! We appreciate your recognition of the effectiveness of our methods. Here is our feedback:
>
> ## More Explanations about Our Methods
> ### RF-Solver
> If we substitute the formulation of derivative into Equation 9, then the formulation becomes
>
> $Z_{t_{i+1}} = Z_{t_{i}}
>     + ({t_{i+1}} - {t_{i}}) v_\theta (Z_{t_{i}}, t_{i})
>     + \frac{1}{2} (t_{i+1} - t_{i})^2 \cdot \frac{v_\theta (Z_{t_{i} + \Delta t}, t_{i} + \Delta t) - v_\theta (Z_{t_{i}}, t_{i})}{\Delta t}.$
>
> In the above formula, $\Delta t$ is required to be set to a **sufficiently small value** to estimate the derivative with less error (as specified in Line 187~188 in paper), which **is not equivalent to** $t_{i+1} - t_{i}$. As a result, RF-Solver exhibits a clear difference from the Heun method. RF-Solver also outperforms Heun method (Table 1).
>
> What's more, RF-Solver is not limited to 2nd order expansion. We derive the **general form** (Equation 7) by firstly deriving the exact formulation of the solution for RF ODE and then applying the high-order Taylor expansion. **This has not been explored by previous works.** If we apply 3rd order expansion, the performance can be further improved ([Table5](https://postimg.cc/4mxRLZNc)).
> ### RF-Edit
> The methods you mentioned are based on U-Net. Among them, Dragdiffusion and Style Inject target point-based editing and style transferring, which is not the core focus of our work. Due to the time limit, we would like to further explore the potential of RF-Edit on these tasks in future work.
>
> Our work focuses on **prompt-based editing using DiT** (such as FLUX and OpenSora). Given DiT's distinct architecture from U-Net and larger parameter count, designing an effective and efficient feature sharing method is non-trivial and underexplored. Addressing this, we thoroughly explore the different choices of feature sharing in DiT (more results are shown in the "More Experiments" Section), proposing RF-Edit. RF-Edit is a unified feature-sharing-based framework which can be applied to various DiT architectures. Achieving satisfying results, we believe RF-Edit is insightful for further work.
>
> ## More Experiments
> Thanks for your valuable and insightful advice! More experiments are provided as follows:
> - Step-vs-quality analysis: Shown in [Table5](https://postimg.cc/4mxRLZNc).
> - Alternative metrics for image editing: Shown in [Table6](https://postimg.cc/XB6kzVb0). At the same time, we would like to kindly point out that LPIPS is also a widely-used metric for measuring the image consistency in previous works of image editing such as PnP (CVPR 2023), Null-Text-Inversion (CVPR 2023).
> - Choices for feature sharing: Further analysis is shown in [Figure11](https://postimg.cc/p5QJNgW5), [Figure12](https://postimg.cc/9rVBPFbd), [Figure13](https://postimg.cc/CdbNffGM), [Figure14](https://postimg.cc/F7ZVT5C2). We finally choose to share the V feature in last 20 single-stream blocks of FLUX in the timesteps near to noise, which is proved to be the most effective choice.
> - Comparisons on more baselines: Results about MasaCtrl are shown in [Table6](https://postimg.cc/XB6kzVb0) and [Figure15](https://postimg.cc/5X2qG7Zj). Results about SANA are shown in [Table8](https://postimg.cc/8F3HLRXc).
> ## Detailed Experiment Setup
> For image editing task, our evaluation dataset consists of about 300 images, following previous works such as PnP (CVPR 2023), InstructPix2Pix (CVPR 2023), Null-Text-Inversion (CVPR 2023) and SmartEdit (CVPR 2024). For the image generation task, we conducted experiments on MS-COCO validation dataset, following previous works such as Stable Diffusion (CVPR 2022). For video inversion and editing tasks, we mainly follow the dataset construction process in COVE (NeurIPS 2024).
>
> For image generation task, the NFE is set to 10 for both RF-Solver and Vanilla RF. The timestep for DPM-Solver++ is set to 20 according to their official github repo. For editing tasks, the hyperparameters for all the baseline methods follow their official github repo for best results. We will add more detailed information in the main paper.
>
> ## Other Questions
> > Doubling baseline solver ... pareto-optimality in lower number of steps.
>
> As illustrated in [Table5](https://postimg.cc/4mxRLZNc), with the increase of timesteps, the performance of both baseline and our methods becomes better. In low timesteps scenarios such as 3 steps for RF-Solver-2 (6 NFE) and 7 steps for Vanilla RF (7 NFE), our methods also illustrate better performance.
> > Discuss potential failure ... computational cost.
>
> The inversion prompt is **optional** and does not significantly impact editing results (you can also see from some examples provided in the supplementary materials) and our method can edit a high-resolution image (1360\*768) with less than 1 minute on a single A100 GPU. The failure case is shown in [Figure16](https://postimg.cc/5XKqM5Vg). We will add these discussions to the main paper！

---

### Decision · Program_Chairs · 2025-05-01

**Decision:**

Accept (poster)

**Comment:**

This paper proposed rectified flow to editing and inversion.

The most of reviewers admits the novelty of this work (eQrG, 1xw9, 5wXG). One contribution is using higher order Taylor expansion to solve ODE resulting in robust preserving unintended mofications (eQrG & 5wXG). Significant experiments support the proposed method (eQrG). On the other hand, reviewers n1SC, 5wXG & eQrG  asked unclear points : dataset selection, solver steps, distinction from baselines, and ablation studies. After rebuttal phase, most of queries from reviewers (5wXG & eQrG) were adequately answered and satisfied by reviewers. Raised concerns from n1SC were not answered properly because of issue with the visibility to authors. However, except the additional queries from n1SC, rebuttals by authors answered previous questions to be made by n1SC and it addressed properly. I believe in resolving the rest of the extra questions in camera-ready process by the authors.

I agreed that this paper will have the positive impact and presents a timely topic to ICML conference. Therefore, I would like to recommend this paper for publication.